# The Clinical Role of CXCR4-Targeted PET on Lymphoproliferative Disorders: A Systematic Review

**DOI:** 10.3390/jcm13102945

**Published:** 2024-05-16

**Authors:** Maryam Zamanian, Domenico Albano, Giorgio Treglia, Alessio Rizzo, Iraj Abedi

**Affiliations:** 1Department of Medical Physics, School of Medicine, Isfahan University of Medical Sciences, Isfahan 81746-73461, Iran; m.zamanian@resident.mui.ac.ir (M.Z.); i.abedi@med.mui.ac.ir (I.A.); 2Nuclear Medicine, ASST Spedali Civili Brescia, 25128 Brescia, Italy; domenico.albano@unibs.it; 3Nuclear Medicine Department, University of Brescia, 25121 Brescia, Italy; 4Faculty of Biomedical Sciences, Università della Svizzera Italiana, 6900 Lugano, Switzerland; 5Division of Nuclear Medicine and Molecular Imaging, Imaging Institute of Southern Switzerland, Ente Ospedaliero Cantonale, 6500 Bellinzona, Switzerland; 6Faculty of Biology and Medicine, University of Lausanne, 1015 Lausanne, Switzerland; 7Department of Nuclear Medicine, Candiolo Cancer Institute, FPO-IRCCS, 10060 Turin, Italy; alessio.rizzo@ircc.it

**Keywords:** lymphoproliferative disorders, lymphoma, leukemia, multiple myeloma, CXCR4, [^68^Ga]Ga-Pentixafor

## Abstract

**Background/Objectives:** We conducted a comprehensive investigation to explore the pathological expression of the CXCR4 receptor in lymphoproliferative disorders (LPDs) using [^68^Ga]Ga-Pentixafor PET/CT or PET/MRI technology. The PICO question was as follows: What is the diagnostic role (outcome) of [^68^Ga]Ga-Pentixafor PET (intervention) in patients with LPDs (problem/population)? **Methods:** The study was written based on the reporting items for systematic reviews and meta-analyses (PRISMA) 2020 guidelines, and it was registered on the prospective register of systematic reviews (PROSPERO) website (CRD42024506866). A comprehensive computer literature search of Scopus, MEDLINE, Scholar, and Embase databases was conducted, including articles indexed up to February 2024. To the methodological evaluation of the studies used the quality assessment of diagnosis accuracy studies-2 (QUADAS-2) tool. **Results:** Of the 8380 records discovered, 23 were suitable for systematic review. Fifteen studies (on 571 LPD patients) focused on diagnosis and staging, and eight trials (194 LPD patients) assessed treatment response. **Conclusions:** The main conclusions that can be inferred from the published studies are as follows: (a) [^68^Ga]Ga-Pentixafor PET may have excellent diagnostic performance in the study of several LPDs; (b) [^68^Ga]Ga-Pentixafor PET may be superior to [^18^F]FDG or complementary in some LPDs variants and settings; (c) multiple myeloma seems to have a high uptake of [^68^Ga]Ga-Pentixafor. Overall, this technique is probably suitable for imaging, staging, and follow-up on patients with LPD. Due to limited data, further studies are warranted to confirm the promising role of [^68^Ga]Ga-Pantixafor in this context.

## 1. Introduction

Lymphoproliferative disorders (LPDs) constitute an assortment of diseases characterized by an inexorable multiplicity of lymphocytes [1]. The uncontrolled regeneration of two subsets of lymphocytes, B and T cells, generates immunoproliferative disorders and predisposes the body to immunodeficiency, a failing immune system, and lymphocyte regulation disruption [2]. Lymphocyte disorders related to each type of B and T lymphocyte can create a heterogeneous set of different disorders [3]. These disorders are known as hematopoietic and lymphatic tumors [4]. LPDs are a pathological hematological condition caused by state pathogens such as infections and unknown oncogenic mutations [3].

LPDs include several diseases, such as different types of leukemia, lymphomas, monoclonal gammopathies (MG), and multiple myeloma (MM). Leukemia is classified as acute lymphoblastic leukemia (ALL) and chronic lymphocytic leukemia (CLL). Also, the type of lymphoma can include types of B and T-cell lymphoma, marginal zone lymphoma (MZL), and mantle cell lymphoma (MCL) [4,5]. These disorders involve the lymph nodes, extra-nodal sites, bone marrow, and blood [5].

Based on a composite reference standard that relies on the type of LPDs, its stage and clinical presentation, and patients’ characteristics (especially age and comorbidities), these diseases are managed by taking advantage of several treatment options, including chemotherapy, immunotherapy, bone marrow transplant, and external beam radiation therapy as single agents or combined [6,7,8,9]. Since the stage at diagnosis is one of the main prognostic factors in most LPDs, precocious diagnosis has a significant impact on treatment and recovery procedures. To date, the most commonly used instruments to determine if a patient has LPD are the blood count and clinical evaluation, and the final diagnosis can be performed only through histopathologic examination [10]. Finally, morphological and metabolic assessments of the disease extent were performed using CT and 2-[^18^F]FDG PET scans, respectively [11].

Through a series of dynamic biological processes, chemoreceptors in cells are helpful for chemotaxis, which is the movement of cells along a chemical cytokine gradient [12]. These receptors include cell surface receptors, which have seven transmembrane domains and are G protein-coupled [13]. C-X-C-motif chemokine receptor 4 (CXCR4) is a member of the chemokine receptor family expressed in immune cells, including B and T lymphocytes, monocytes, macrophages, neutrophils, and eosinophils. Furthermore, it is extensively expressed in cancer and metastases. The pathological expression of the CXCR4 receptor has been observed in various cancers, including lung, breast, pancreatic, prostate, and hematologic malignancies [14,15]. Overexpression of the CXCR4 receptor and its activation by CXCL12 (its ligand) binding contribute to tumor growth and metastasis [16,17]. Its increased expression may suggest resistance to chemotherapy and poor prognosis. These observations made this receptor a suitable target for molecular imaging, both in a staging setting and for the assessment of treatment response, as well as for theragnostic [15,18].

Molecular imaging approaches relying on the expression of this receptor have recently been explored, particularly for detecting leukemia [19], and several labeling techniques have been tested for imaging to study the expression level of the CXCR4 receptor. In this context, several radiopharmaceuticals labeled [^68^Ga]Ga, [^18^F]F, [^64^Cu]Cu, and [^111^In]In have been developed [20,21,22,23].

Pentixafor (or CPCR4.2) is a pentapeptide structure with a high-affinity nuclear probe based on the cyclo (D-Tyr1-[NMe]-D-Orn2-Arg3-2-Nal4-Gly5) scaffold coupled with DOTA via 4-(aminomethyl) benzoic acid [24]. [^68^Ga]Ga-Pentixafor (a radio-labeled CXCR4-ligand) was released in 2011 for PET imaging, promising a new era in diagnosis and therapy [25]. Furthermore, this tracer has been evaluated using PET/CT and PET/MRI to study CXCR4 receptor expression in various lymphoma subtypes and leukemias [26,27,28,29,30,31].

In the present systematic review, we assessed the potential of [^68^Ga]Ga-Pentixafor PET to investigate all types of LPDs from multiple perspectives of diagnosis, staging, and therapy response in a systematic study.

## 2. Materials and Methods

### 2.1. Protocol and Registration

The present research aimed to examine the role of [^68^Ga]Ga-Pentixafor PET, PET/CT, and PET/MRI in evaluating LPDs. The study protocol was composed following the reporting items for systematic reviews and meta-analyses (PRISMA) 2020 guidelines, provided in Appendix A [32]. Meanwhile, the study was registered on the prospective register of systematic reviews (PROSPERO) website (CRD42024506866).

The search question, based on the problem, intervention, control, and outcome (PICO) model, was as follows: What is the diagnostic role (outcome) of [^68^Ga]Ga-Pentixafor PET (intervention) in patients suffering from LPDs (problem/population) compared to other imaging examinations (control)?

### 2.2. Literature Review Method and Study Selection

The following four electronic databases were examined: Scopus, MEDLINE (Pub-Med), and Embase (Elsevier and Ovid). The Scholar database was used for a grey search. After selecting the eligible studies, their references were screened to find more studies suitable for inclusion. The string search strategy included the following words: (“Pentixafor” OR “CXCR4”) AND (“lymphoproliferative” OR “lymphoma” OR “myeloma” OR “leukemia” OR “leukemia”) AND (“PET” OR “Positron Emission Tomography”). Furthermore, the clinical trial.gov website was examined in the process (condition/disease: lymphoproliferative disorders; other terms: PET OR positron emission tomography; intervention/treatment: Pentixafor OR CXCR4).

The study search was conducted in December 2023, applying a time frame filter between 2010 and 2023 (December), and updated in 2024 (February 27). Other filter results through “title, abstract, and keywords” in the Scopus database, in addition to the time filter, were included regarding document type “Article”, source type “journal”, in the Embase database using quick search was publication type “Article”.

Two people performed the initial search, which was conducted using the Endnote software (Version 21, USA) to accurately screen each title, abstract, and full-text evaluation step. The steps of data extraction and study selection, data extraction, and quality assessment were carried out by the same two people. The overlap between their results was 99%, and in a meeting regarding their difference, a conclusion was made based on the include and exclude criteria. Again, the search was revised by one person for the possibility of missing studies on the date (March 2024) and its results were added to the study.

### 2.3. Inclusion and Exclusion Criteria

The inclusion criteria were studies analysing LPDs using [^68^Ga]Ga-Pentixafor PET, PET/CT, or PET/MRI.

The exclusion criteria were (a) studies that were reviews, conference abstracts, case reports, small case series editorials, letters, and notes; and (b) studies that were outside the field of interest of this review, such as other tracers and equipment.

### 2.4. Data Extraction

The features of the study are presented in two tables. The first category included descriptive specifications (author name, publication year, country, study type, sample size, sex, age, and lymphoproliferative disorder type). The second was technical characteristics, including aim, device, dose injection, uptake time, semi-quantitative parameters, organ involvement, standard reference, and device model.

### 2.5. Quality Assessment (Risk of Bias)

One person performed the methodological evaluation of the studies using the quality assessment of diagnosis accuracy studies-2 (QUADAS-2) tool. The quality assessment levels of this tool were low, unclear, and high, respectively.

## 3. Results

### 3.1. Collected Studies and Quality Assessment

This systematic review eventually included 23 of the 8380 records that were initially identified. The first screening round was accomplished by deleting duplicates based on research titles using Endnote (21-Version, Thompson Reuters, Toronto, ON, Canada). For studies of doubtful type and accessibility, the abstract and full text were used as the next screening stage. The inclusion and exclusion criteria guided the selection and exclusion of the study. The results of the reference screening were similar to those of the search results, and no additional studies were included in the review. None of the published articles on the ten protocols registered on clinicaltrial.gov were available. Figure 1 shows the selection steps based on the PRISMA flowchart.

The methodological quality of the studies that were part of the evaluation, based on the criteria of the QUADAS-2 tool, was low, unclear, and high. The assessment was initially recorded using Excel software (Microsoft 2021, 365 subscription, Version 2404 Build 16.0.17531.20140), as shown in Figure 2.

The highest risk of bias related to the patient selection criteria was approximately 60%. Since most of the studies were exploratory and had a small number of patients, random selection of patients and division of the control group were not considered.

In addition, the risk of bias for the index test and the reference standard was approximately almost equal and about 17%. The lowest risk of bias was observed for flow and timing (about 5%).

### 3.2. Description of Studies

A total of 23 studies qualified for systematic examination. This research looked at the use of [^68^Ga]Ga-Pentixafor PET/CT or PET/MRI technology in various forms of LPD. Eleven studies were prospective, whereas eight were retrospective. These investigations were published between 2016 and 2023. Seven studies were conducted in Austria, five in Germany, five in China, three in Turkey, one in India, one jointly between Germany and the USA, and one between Germany and Austria.

Most of the study’s findings included diagnostic procedures for lymphoma and MM. This study included 765 participants with LPD. Fifteen studies focused on diagnosis and staging, and eight trials assessed the treatment response. In the 12 studies, the results of [^68^Ga]Ga-Pentixafor and [^18^F]FDG PET, PET/CT, or PET/MRI were compared for all types of LPDs.

Table 1 and Table 2 provide an overview of the descriptive and clinical features of the chosen studies.

### 3.3. Data Clustering

The substantial degree of research heterogeneity made meta-analysis impossible. We then performed a systematic evaluation of the following subgroups: MM, MZL/MALT, MCL, WM/LPL, CNSL, and CLL/ALL.

#### 3.3.1. MM

Diagnostic accuracy: By examining patients affected by LPDs from two centers, Buck et al. found that the highest mean SUVmax and TBR indices were related to MM disorders [36].

Diagnostic accuracy—Comparison with [^18^F]FDG: Examining of patients who had already been treated with various chemotherapy drugs in Lapa et al.’s study showed that the results of [^68^Ga]Ga-Pentixafor PET/CT were positive for 23/35 patients. In total, 8/23 patients had BM intramedullary involvement, 13/23 individuals had both intra—and extramedullary involvement, and 2/23 patients had only extramedullary involvement. Most extramedullary involvement was in the soft tissue, and intramedullary was in the appendicular skeleton. Of the 19 patients who underwent both [^18^F]FDG and [^68^Ga]Ga-Pentixafor scans, 11/19 were CXCR4-positive and 4/19 were CXCR4-negative. CXCR4 positivity in PET scans was recognized as a negative prognostic factor [50]. By evaluating ten patients, Zhou et al. showed abnormal [^18^F]FDG PET/CT uptake in all patients and [^68^Ga]Ga-Pentixafor uptake in 50% of patients. There was a significant correlation between the infiltration of the involved lumbar vertebrae (L2–L4) and SUVmean and TBRmean for [^68^Ga]Ga-Pentixafor, whereas this correlation was not calculated for [^18^F]FDG [43]. Furthermore, some studies confirmed the better detection rate of [^68^Ga]Ga-Pentixafor PET/MRI compared to [^18^F]FDG in detecting MM; these findings were compatible with laboratory biomarkers and organ damage [48,50]. Pan et al. found a higher positive rate for [^68^Ga]Ga-Pentixafor PET/MRI in 30 newly diagnosed patients [48]. Ozkan et al., by examining 24 patients, showed that 13/24 (54%) and 14/24 (67%) focal pathological lesions were detected by [^68^Ga]Ga-Pentixafor PET/CT and [^18^F]FDG techniques, respectively. Also, the results were concordant in 19/24 patients [46].

Impact on staging—Comparison with [^18^F]FDG: Shekhawat et al. compared [^68^Ga]Ga-Pentixafor and [^18^F]FDG PET/CT in a staging setting; [^68^Ga]Ga-Pentixafor was able to change the stage in 14/34 (41.0%) patients compared to conventional functional imaging. [^18^F]FDG, compared to [^68^Ga]Ga-Pentixafor, had inferior diagnostic performance in two patients and worked similarly in nine. Furthermore, the maximum tumor-to-background ratio (TBR)max of [^68^Ga]Ga-Pentixafor was higher than that of [^18^F]FDG in 27/34 patients and lower in 6/34 patients, and none of the two tracers were absorbed in one patient (*p* < 0.001). Both techniques were used to detect extramedullary lesions in six patients; in contrast, medullary lesions were detected in five individuals by [^68^Ga]Ga-Pentixafor and in three patients by [^18^F]FDG PET. The percentage of BM plasma cells and TBRmax were the only variables that showed a significant association (*p* = 0.013). The [^68^Ga]Ga-Pentixafor technique upstaged more patients than [^18^F]FDG PET imaging [26].

Prediction of treatment response—Comparison with [^18^F]FDG: Kraus et al., by evaluation of 87 MM with [^68^Ga]Ga-Pentixafor PET/CT scans, demonstrated heterogeneity in spleen uptake. However, there was a direct inverse relationship between the uptake intensity and disease progression. Enrolled patients’ overall survival (OS) was significantly shorter in subjects with lower uptake, expressed as SUVpeak [35]. Kuyumcu et al. reported that in 24 patients, [^68^Ga]Ga-Pentixafor and [^18^F]FDG PET/CT were positive in 17 and 13 patients; also, the positive rate was 70.8 and 54.1%, respectively. Total bone marrow (TBM) uptake of both [^68^Ga]Ga-Pentixafor and [^18^F]FDG was related to OS, serum beta-2-microglobulin levels, and the CRAB (CRP + albumin) score [30].

#### 3.3.2. MZL/MALT

Diagnostic accuracy: When Haug et al. used [^68^Ga]Ga-Pentixafor PET/MRI to investigate MZL and B-cell lymphoma of the mucosa-associated lymphoid tissue (MALT) lymphoma, they observed high radiopharmaceutical uptake in all of the regions affected, including the stomach, orbit, lungs, and other involved areas in most patients (33/36). Three patients with no increase in uptake had previous orbital MALT lymphoma surgery [47]. Other results of examining patients with LPD in the study by Buck et al. used PET/CT, as the SUVmax value was in descending order for MZL (12), B (10), and T-cell lymphoma (8). The TBR values were the same as those for MZL (6.9), B (6.2), and T-cell lymphoma (5.7) [36]. Kosmala et al. examined the tumor burden in 73 MZL patients by [^68^Ga]Ga-Pentixafor PET/CT and showed a decrease in uptake in the lymphoma-sink effect regions, which occurred as decreased uptake in the normal organs of patients with a high-burden tumor status [34].

Diagnostic accuracy—Comparison with [^18^F]FDG: Kuyumcu et al. showed that the positive rate in [^68^Ga]Ga-Pentixafor PET/CT was 4/5 for B-lymphoma and 1/5 for [^18^F]FDG and 3/4 for T-lymphoma and 1/4 for [^18^F]FDG. So, the uptake rate of [^68^Ga]Ga-Pentixafor is significantly higher. The results of the evaluation of patients with MALT lymphoma in this study showed an unusual type of lymphoma. It is a rare type of MALT disorder that causes increased lung uptake and metabolic activity [37].

Impact on staging: Duell et al. reported that [^68^Ga]Ga-Pentixafor PET/CT performed better than conventional staging methods in all 22 newly diagnosed MZL patients (*p* < 0.001); in most cases (16/18 patients), the results were confirmed by PET-guided biopsy. Generally, one-third of the patients in the treatment protocols and half of the stages were affected by the application of this approach [42].

Prediction of treatment response: Mayerhoefer et al. tried finding an alternative to the biopsy follow-up after eradicating Helicobacter pylori using [^68^Ga]Ga-Pentixafor PET/MRI. They examined 26 gastric MALT lymphomas and 20 control patients without lymphoma. The accuracy, sensitivity, and specificity results at follow-up were 97.0%, 95.0%, and 100.0%, and positive and negative predictive values were 100.0% and 92.9%, respectively. Moreover, high TBR indices were reported in the semiquantitative analysis [29].

#### 3.3.3. MCL

Diagnostic accuracy: Mayerhofer et al. examined the ability of this method to assess treatment response. The results of [^68^Ga]Ga-Pentixafor PET/MRI were compared with those of MRI. Lesion uptake was high in the beginning, and during follow-up, [^68^Ga]Ga-Pentixafor and MRI were able to estimate complete remission lesions in about 70.2% (40/57) and 47.4% (27/57) of patients, respectively (*p* = 0.021). None of the patients with complete remission confirmed by [^68^Ga]Ga-Pentixafor showed progression in the subsequent follow-up, confirming the results of this diagnostic technique [33]. Finally, Buck et al. reported that the uptake in MCL lymphoma (20 patients) was the highest (SUVmax 14) after MM. Similarly, the TBR value was the highest after MM (TBR 9) [36].

Diagnostic accuracy—Comparison with [^18^F]FDG: In a study by Kuyumcu et al., patients showed different uptake results of Pentixafor PET/CT. In the first patient, there was no significant uptake of [^68^Ga]Ga-Pentixafor in [^18^F]FDG-positive lymph nodes; in the second patient, [^68^Ga]Ga-Pentixafor uptake was higher in the positive BM and abdominal lymph nodes, and in the third patient, heterogeneous uptake was observed in multiple cervical lymph nodes [37]. Mayerhoefer et al. analyzed 22 MCL patients in stages I–IV, which involved 109 anatomical regions in nodal (99/109) and extra-nodal sites except for the BM and spleen (10/109), including the Waldeyer ring, stomach, lung, and soft tissues. The BM was also involved in 11 patients. In this study, the [^18^F]FDG PET/MRI technique was used as a reference standard to evaluate the results of [^68^Ga]Ga-Pentixafor PET/MRI. The agreement between the results of the two techniques was 74.6%. [^68^Ga]Ga-Pentixafor SUVmax and SUVmean uptake were significantly higher than [^18^F]FDG in all patients. The number of false-positive regions for [^68^Ga]Ga-Pentixafor was seven versus two regions for [^18^F]FDG; therefore, the PPV value was 94.0% versus 96.5%. The SUV and TBR indices were higher in the patients with BM and spleen involvement. They also demonstrated a much greater diagnostic sensitivity of 100% versus 75.2% for [^18^F]FDG (*p* < 0.001) [38].

#### 3.3.4. WM/LPL

Prediction of treatment response—Comparison with [^18^F]FDG: Pan et al., using [^68^Ga]Ga-Pentixafor PET/CT on 15 WM/LPL patients, showed the involved regions, including BM in all patients (15/15) and lymph nodes in most patients (12/15), had a significant increase in uptake. Furthermore, three patients had the para-medullary disorder, one patient had involvement of the thoracic and sacral nerve roots, two patients had focal hepatic or pancreatic lesions, and two patients had spleen involvement. [^68^Ga]Ga-Pentixafor showed abnormal uptake in all involved extra-medullary regions, whereas [^18^F]FDG was positive in only some areas. In addition, [^18^F]FDG showed increased uptake in the involved BM only in the number of patients (15/10) and could not detect additional lesions. Follow-up evaluations were conducted two weeks to ten months after the last chemotherapy session. In the five patients (with complete remission), normal uptake of BM and extramedullary regions using [^68^Ga]Ga-Pentixafor was assessed as decreased; however, the uptake of [^18^F]FDG in the four patients decreased from a score of (3–4) to a score of 2 and did not change significantly in the fifth patient. In other patients, [^68^Ga]Ga-Pentixafor did not show any significant change with various degrees of reduction in [^18^F]FDG uptake [41]. In another study that was conducted—again by Pan et el.—on a similar number of patients, the [^68^Ga]Ga-Pentixafor PET/CT uptake was significantly higher than that of [^18^F]FDG in the affected patients, and in addition to BM, lymph nodes also showed high uptake [40].

Luo et al. examined the higher rate of positive results in 17 patients for BM, lymph nodes, and other extramedullary involvements. They observed that [^68^Ga]Ga-Pentixafor PET/CT scanning had a higher positive rate than [^18^F]FDG PET/CT (100% vs. 58.8%; *p* = 0.023), and, regarding lymph nodes, the rates were 76.5%, and 11.8%, respectively. The sensitivity of [^68^Ga]Ga-Pentixafor PET/CT and [^18^F]FDG PET/CT scanning to evaluate BM involvement was 94.1% and 58.8%, respectively. In addition, [^68^Ga]Ga-Pentixafor showed more para-medullary and CNS involvement [31].

#### 3.3.5. CNSL

Diagnostic accuracy: Herhaus et al. evaluated 11 CNSL with primary and secondary involvement stages, resulting in B cell lymphoma, and visual evaluation with [^68^Ga]Ga-Pentixafor PET/CT was positive in 10/11 cases. In addition, the SUV and TBR indices of the involved regions were relatively high, and the only patient who did not show high uptake had previously undergone surgery. The expression of CXCR4 in nine patients was found to be high. There was a correlation between the expression of CXCR4 and treatment response, as lower [^68^Ga]Ga-Pentixafor uptake revealed a better response to treatment [44]. In addition, Starzer et al. evaluated five patients and reported that [^68^Ga]Ga-Pentixafor PET/MRI could detect lesions in 5/7 patients. The correlation between the ADCmean and SUV indices and TBR was moderate. In addition, the results of changes in planning target volume (PTV) as a treatment response evaluation using [^68^Ga]Ga-Pentixafor PET/MRI showed that follow-up and progression were very similar to the results of contrast-enhanced MRI [39].

Diagnostic accuracy—Comparison with [^18^F]FDG: In a study by Chen et al., 26 patients with diffuse large B-cell lymphoma had CNS lesions. No false positive lesions were found using [^68^Ga]Ga-Pentixafor PET/CT; therefore, its accuracy was 100%. Increased uptake in the bilateral lateral ventricle and corpus callosum regions indicated involvement. Although the SUVmax of [^68^Ga]Ga-Pentixafor was lower than that of [^18^F]FDG, the T/N ratio was significantly higher in most of the patients. CXCR4 expression in lesions was 74%, and there was a significant correlation between that and SUVmax in all lesions [27].

#### 3.3.6. CLL/ALL

Diagnostic accuracy: By evaluating nine patients with CLL after treatment with ibrutinib, Mayerhoefer et al. showed a decrease in BM and lymph node uptake but an increased [^68^Ga]Ga-Pentixafor PET/CT; (PET/MRI) accumulation in the spleen was associated with increased leukocytosis. In the long-term treatment of 2–4 months, similarly, the SUVmean in the BM and lymph nodes decreased, but the increasing trend in the SUVmean in the spleen was reduced or eliminated. Long-term follow-up (more than six months) after the start of treatment showed a decrease in leukocytosis along with a decrease in BM SUVmean and mean and volume of the lymph nodes and spleen compared to the beginning [45]. In addition, after the evaluation of ten CLL with 20 controls (including ten pancreatic adenocarcinoma and ten MALT patents) by PET/CT, the SUVmax and SUVmean values for the pelvis and L4 were observed between the CLL and the two control groups. In contrast, only a significant difference was observed between the two control groups in the SUVmean at L4. The results of statistical tests showed that the SUVmax and SUVmean of lymph nodes involved in eight CLL patients were among the hottest lesions of the BM pelvis and L4. There was no significant correlation between SUVmean and ADCmean at the exact anatomical locations of the pelvis. In addition, there was no significant correlation between SUVmax and SUVmean of the BM and the leukocyte count, lymphocyte percentage, lactate dehydrogenase, β2-microglobulin, and C-reactive protein (*p* > 0.05). However, there was a significant negative correlation between the ADCmean values in the pelvis and both the leukocyte count (r = −0.78; *p* = 0.002) and the lymphocyte percentage (r = −0.81; *p* = 0.001) [49].

In a study by Buck et al., the SUVmax for the two types of CLL and ALL were almost close to each other at 11.6, which was lower than that of MM, MZL, and MCL. However, the TBR index between the two types was different and equal to 8.2 for CLL and less than 7 for ALL [36].

Diagnostic accuracy—Comparison with [^18^F]FDG: Kuyumcu et al., in a comparative study using PET/CT, showed that in the first patient, the lymph nodes showed no significant uptake, but BM involvement was evident. In the second patient, the pelvic hypermetabolic mass and abdominal lymph nodes show much more using [^18^F]FDG, which means the disease progression with the CXCR4 expression is low or mild [37].

## 4. Discussion

Examining the expression level of CXCR4 has recently been considered for the development of a PET-based imaging technique and subsequent quantitative evaluation of blood disorders and tumors, including LPDs since this receptor is overexpressed in various blood cells and solid tumors [51,52]. Increased CXCR4 receptor expression in LPDs is the rationale for using [^68^Ga]Ga-Pentixafor for PET in different diagnostic settings [22]. This molecular imaging method, which includes PET, PET/CT, and PET/MRI, is a suitable diagnostic technique for distinguishing the active state of the disease from the inactive state. To the best of our knowledge, no comprehensive review has been conducted to investigate all types of LPDs using [^68^Ga] Ga-Pentixafor for PET to estimate its ability from different aspects.

According to the findings, most patients with lymphoma, MM, and leukemia types often presented higher SUV and TBR values during imaging using [^68^Ga]Ga-Pentixafor. Abnormal radiopharmaceutical absorption can be observed in the bone marrow, lymph nodes, extra-nodal sites (including the liver and lungs), and spleen. The higher TBR value reported in the included studies, as well as providing a quantitative scale that can be evaluated and compared, is not ineffective in delivering a higher-quality image in [^68^Ga]Ga-Pentixafor for PET scans. The results of this index correspond to BM plasma filtration. All instruments employed, including PET, PET/CT, and PET/MRI, showed this difference [27,33,36,38,43,45]. Quantitative determination of the expression level of CXCR4 can also differentiate between LPD disorders [36,51]. Furthermore, BM involvement has often been detectable only with [^68^Ga]Ga-Pentixafor in patients with CLL [37].

[^68^Ga]Ga-Pentixafor PET showed an overall higher positive rate than [^18^F]FDG PET in LPDs, which confirms its greater detection rate in the evaluation of this disorder [38,39]. Evaluations showed that the metabolic approach ignored the involvement of some lymph nodes. As MM is compared to other LPDs, the absorption rate of [^68^Ga]Ga-Pentixafor is higher, which is probably caused by the high expression of the CXCR4 receptor and the high involvement of the body, including the BM and blood [26,48,50].

Another advantage of using [^68^Ga]Ga-Pentixafor PET is CNSL. The site of growth of this type of lymphoma, radiation treatment, and central nervous system characteristics make evaluating and assessing the response to treatment challenging [53]. The results obtained from [^68^Ga]Ga-Pentixafor for PET were consistent with the results of CE-MRI [27]. Complete remission of lymphoma during treatment is challenging to detect with MRI alone, even with contrast, because of non-enhancing lesions and wide-spread disease, and scar tissue formed in the treatment area due to radiation hampers distinguishing it from the diseased area with MRI [54,55]. However, examination using PET/MRI equipped with diffusion-weighted imaging (DWI) can solve this problem because it can differentiate necrotic tissue from recurrent lymphoma [39]. In addition, some patients are prohibited from administering contrast or face restrictions because of pacemakers [56]. Furthermore, the use of [^18^F]FDG to diagnose and track therapy response in the brain is difficult due to its high uptake in normal brain tissues. Likewise, patients do not need to fast to receive [^68^Ga]Ga-Pentixafor, in contrast to [^18^F]FDG, and greater contrast is produced using [^68^Ga]Ga-Pentixafor [55].

It has also been confirmed that [^68^Ga]Ga-Pentixafor is superior to metabolic PET imaging for staging MM and MZL disorders [26,43]. Its high accuracy in assessing treatment response and patient follow-up has been reported and compared to [^18^F]FDG, using biopsy results and laboratory test values for some types of LPDs, including MZL, MCL, CNSL, WM, CLL/ALL, and MM [29,33,39,40,41,45]. The relationship between the expression of CXCR4 and the decrease in SUV indices on CLL, ALL, and MM, and the OS can help predict disease recurrence and prognosis [30,35,45].

The agreement between the [^68^Ga]Ga-Pentixafor PET findings and clinical results regarding the response to treatment suggests that this technique is also suitable for therapy response assessment. This approach seems to be able to distinguish between LPD and other cancers.

Comparison between different modalities used, such as PET/CT and (/MRI), can provide us with good diagnostic information, but this goal faces several limitations. First, in different lymphoproliferative studies, the number of studies in different modalities is either low or we only have one type of modality. In addition, few studies have reported the sensitivity and specificity of the modalities, and often, the same evaluation index is not found in different groups, which is a requirement for scientific comparison.

To date, only one systematic review by Albano et al. (in 2022) assessed the role of [^68^Ga]Ga-Pentixafor PET in lymphomas [57]. The current study is broader, including patients with LPDs other than lymphomas. We found increased [^68^Ga]Ga-Pentixafor uptake in MM and confirmed that radiopharmaceutical uptake was higher in MCL and MZL than in other B- and T-cell lymphomas.

The significant differences in the protocol approach, such as the injected dosage tracer, uptake time, different device scanners, and reconstruction design parameters, were a limitation of the studies and a barrier to the implementation of the meta-analysis. The second limitation was the small number of studies examining the ability of [^68^Ga]Ga-Pentixafor to stage LPDs and the small sample size. Third, there are insufficient standard references and biopsies to assess bone and lymphatic involvement. The last limitation is the probability that chemotherapy medications affect the expression of CXCR4 receptors.

Although most of the findings have demonstrated the efficacy of the [^68^Ga]Ga-Pentixafor PET, additional research on the expression level of CXCR4 receptors in different types of LPDs and their staging in larger sample sizes is required. Furthermore, by conducting more studies, we hope to diagnose the type of LPD disorder based on the expression level of CXCR4.

## 5. Conclusions

In general, it can be concluded that the [^68^Ga]Ga-Pentixafor PET is suitable for the diagnosing, staging, and follow-up of patients with LPD disorders. It appears to be a complementary technique to [^18^F]FDG PET in several clinical scenarios. However, the usefulness of this method needs to be further investigated for different types of LPD.

## Figures and Tables

**Figure 1 jcm-13-02945-f001:**
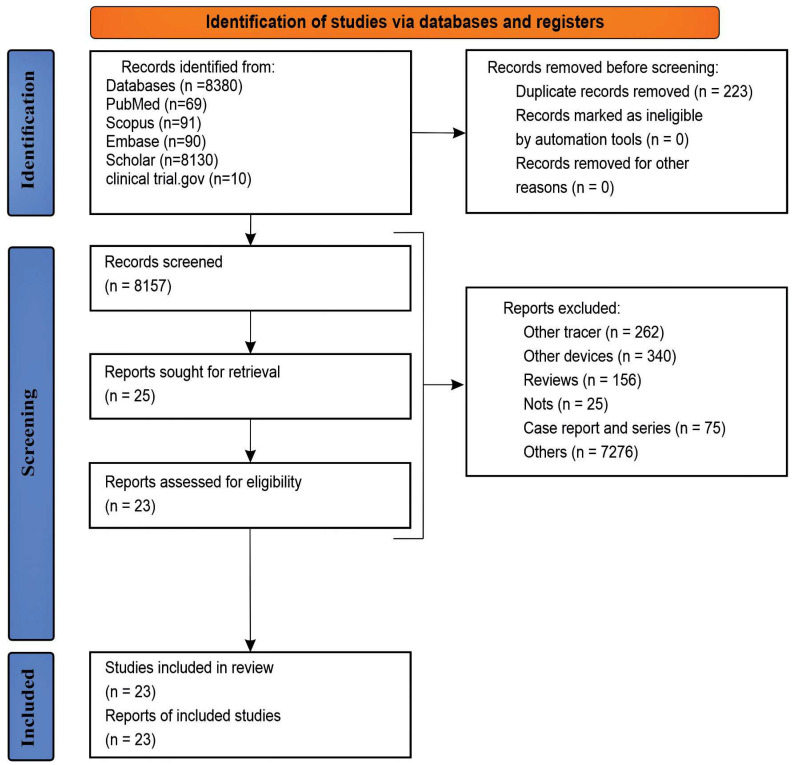
2020 PRISMA flowchart of study selection for the systematic review.

**Figure 2 jcm-13-02945-f002:**
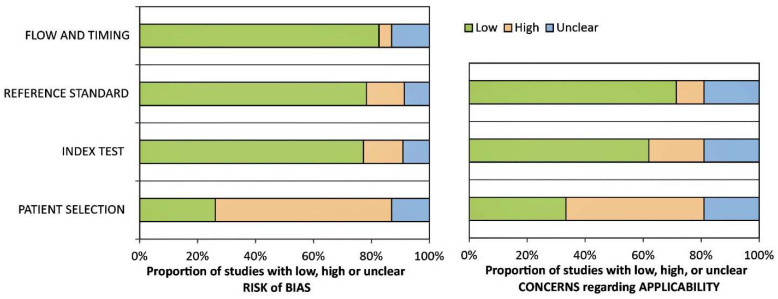
The overall results of the quality assessment evaluation of studies included in the systematic review by using the QUADAS-2 tool.

**Table 1 jcm-13-02945-t001:** Descriptive specifications from selected studies.

First Author	Publication Year	Country	Sample Size	Sex	Age Mean (or Min–Max) in Years	Study Type	Lymphoproliferative Disorder Type
Mayerhoefer et al. [33]	2023	Austria	16	7F 9M	69.9 ± 7.9	Prospective	MCL
Kosmala et al. [34]	2023	Germany and USA	73	NM	66.2 ± 12.3	Retrospective	MCL
Shkhawat et al. [26]	2022	India	34	13F 21M	57.5 (35–78)	Retrospective	MM
Kraus et al. [35]	2022	Germany	87	34F	59 ± 8	NM	MM
Buck et al. [36]	2022	Germany and Austria	389	NM	NM	Retrospective	MCL (20); MZL (187); MM (113); B-cell lymphoma (10); T-cell lymphoma (3); CLL (50); ALL (6)
Chen et al. [27]	2022	China	26	8F 18M	NM	Retrospective	CNSL
Mayerhoefer et al. [29]	2022	Austria	26	12F 14M	64.1 ± 11.9	Prospective	MALT
Kuyumcu et al. [37]	2021	Turkey	11	4F 7M	56.8	Retrospective	CLL (2) B-cell lymphoma (5) T-cell lymphoma (4)
Mayerhoefer et al. [38]	2021	Austria	22	11F 11M	70.0 ± 8.5	Prospective	MCL
Starzer et al. [39]	2021	Austria	7	4F 3M	54.9 ± 15.0	Prospective	CNSL
Pan et al. [40]	2021	China	15	3F 12M	60.9 ± 8.6	Prospective	WM
Pan et al. [41]	2021	China	15	3F 12M	60.9 ± 8.6	Prospective	WM
Kuyumcu et al. [30]	2021	Turkey	17	11F 6M	60.2 ± 11.5	Retrospective	MM
Duell et al. [42]	2021	Germany	22	15F 7M	65	Retrospective	MZL (MALT)
Zhou et al. [43]	2020	Germany	10	2F 8M	62	Retrospective	MM
Herhaus et al. [44]	2020	Germany	10	2F 8M	62	Retrospective	MM
Mayerhoefer et al. [45]	2020	Germany	11	3F 8M	64.1	Retrospective	CNSL
Luo et al. [31]	2020	Austria	9	NM	NM	Prospective	CLL
Ozkan et al. [46]	2019	China	17	6F 11M	62.6 ± 10.5	Prospective	WM/LPL
Haug et al. [47]	2019	Turkey	24	17F 7M	58.9 ± 12.7	NM	MM
Pan et al. [48]	2019	Austria	36	19F 17M	62	Prospective	MALT
Mayerhoefer et al. [49]	2020	China	30	11F 19M	59.1 ± 9.8	Prospective	MM
Lapa et al. [50]	2018	Austria	13	6F 7M	65.6 ± 11.2	Prospective	CLL

NM: not mentioned; F: female; M: male; MBq: mega Becquerel; MZL: marginal zone lymphoma; MCL: mantle cell lymphoma; CNSL: central nervous system lymphoma; CLL: chronic lymphocytic leukemia; ALL: acute lymphoblastoid leukemia; MM: multiple myeloma; WM: Waldenström macroglobulinemia; LPL: lymphoplasmacytic lymphoma.

**Table 2 jcm-13-02945-t002:** Technical characteristics from the selected studies.

First Author	Aim	Device	Injected Activity (MBq)	Uptake Time (min)	Semiquantitative Parameters	Target Involvement	Standard Reference	Comparison with [^18^F]FDG PET
Mayerhoefer et al. [33]	Treatment response assessment	PET/MRI	150 MBq	60	SUVmax SUVmean	Liver, and blood pool uptake	Biopsy	No
Kosmala et al. [34]	Diagnosing	PET/CT	128 ± 24.5 MBq	60	SUVmean SUVmax SUVpeak	Nodal, extra-nodal, heart, liver, spleen, bone marrow, kidneys	Biopsy and expert reader	No
Shkhawat et al. [26]	Staging	PET/CT	148–185 MBq	60	SUVmax TBRmax	Bone marrow	Laboratory test	Yes
Kraus et al. [35]	Treatment response assessment	PET/CT	43–207 MBq	NM	SUVpeak TBR	Splenic	Laboratory tests and biopsy	No
Buck et al. [36]	Diagnosing	PET/CT	134 MBq	60	SUVmax SUVmean SUVpeak TBR	NC	Expert reader	No
Chen et al. [27]	Diagnosing	PET/CT	77.7–166.5 MBq	60	SUVmax	CNS lesion	Biopsy, T/N, histology, and the MRI	Yes
Mayerhoefer et al. [29]	Treatment response assessment	PET/MRI	150 MBq	60	SUVmax SUVmean TBR	Gastric	Biopsy, immunohistochemistry	No
Kuyumcu et al. [37]	Diagnosing	PET/CT	185 MBq	60	SUVmax	Bone marrow, lymphnodal, extra-lymphnodal,	Biopsy and histology	Yes
Mayerhoefer et al. [38]	Diagnosing	PET/MRI	150 MBq	60	SUVmax SUVmean TBR	Nodal, and extra-nodal in Waldeyer ring; lungs; liver; pancreas; stomach; small intestine; large intestine; adrenal glands; kidneys; soft tissues (skin/fat/muscle) and other	MRI and biopsy	Yes
Starzer et al. [39]	Treatment response assessment	PET/MRI	150 MBq	60	SUVmax SUVmean VOL_MRI_	CNS lesion	Histology	No
Pan et al. [40]	Treatment response assessment	PET/CT	85.1 ± 27.4 MBq	45.9 ± 19.7	MTV TLU_CXCR4_ SUVpeak	Bone marrow	Laboratory tests and biopsy	Yes
Pan et al. [41]	Treatment response assessment	PET/CT	2.8/kg	56	SUVmax TBRbloodTBRliver	Blood, liver, Nodal, extra-nodal	Laboratory tests and biopsy	Yes
Kuyumcu et al. [30]	Treatment response assessment	PET/CT	185 ± 10 MBq	60	SUVmax, SUVmean TMB	Bone marrow	Clinical, laboratory test findings, imaging, histology	Yes
Duell et al. [42]	Staging and Treatment response assessment	PET/CT	117 MBq	60	SUVmax	gastrointestinal tract, bone marrow, lymph nodes, tonsils, glands, soft tissues	Biopsy and histology	No
Zhou et al. [43]	Diagnosing	PET/CT	2–3 MBq/kg	60	SUVmax SUVmean TBRmax TBRmean	Bone marrow	Laboratory tests and biopsy	Yes
Herhaus et al. [44]	Diagnosing	PET/CT	2–3 MBq/kg	60	SUVmax SUVmean TBRmax TBRmean	Bone marrow	Laboratory tests and biopsy	Yes
Mayerhoefer et al. [45]	Diagnosing	PET/CT and PET/MRI	1–2.9 MBq/kg	NM	SUVmax TBR	CNS lesion	Biopsy, histology, and expert reader	No
Luo et al. [31]	Treatment response assessment	PET/MRI	150 MBq	60	SUVmax, SUVmean PTV	Lymph nodes, bone marrow, spleen	Biopsy	No
Ozkan et al. [46]	Diagnosing	PET/CT	84.6 ± 26.2 MBq	30–90	SUVmax	Bone marrow; lymph node; CSF	Bone marrow aspiration and biopsy	Yes
Haug et al. [47]	Diagnosing	PET/CT	130–185 MBq	30	SUV	Bone marrow and kidney	Laboratory tests and expert reader	Yes
Pan et al. [48]	Diagnosing	PET/MRI	172 MBq	60	SUVmax SUVpeak SUVmean TBRmean	Orbit; stomach; lungs; soft tissues; adrenal gland; tonsils; parotid gland; and urinary bladder	Endoscopy, biopsy, and histology	Yes
Mayerhoefer et al. [49]	Diagnosing	PET/CT	40.7–170.2 MBq	30–90	SUVmean, SUVmax TBmV	Bone marrow	Expert reader	Yes
Lapa et al. [50]	Diagnosing	PET/MRI	150 MBq	60	SUVmax, SUVmean	Bone marrow and spleen	Laboratory test	No

NC: not cleared; SUV: standardized uptake value; max: maximum; TBR: tumor-to-background ratio; OS: overall survival; TTP: time-to-progression; T/N: tumor to normal brain; VOL: volume.

## Data Availability

Not applicable.

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
