# Peer review of "The Clinical Role of CXCR4-Targeted PET on Lymphoproliferative Disorders: A Systematic Review"

_jcm, 2024, doi:10.3390/jcm13102945_

Round 1

Reviewer 1 Report

Comments and Suggestions for Authors

The literature review provides a comprehensive evaluation of 68Ga-pentixafor PET/CT imaging in lymphoproliferative disorders (LPDs), demonstrating its potential as a valuable diagnostic tool. It effectively summarizes the utility of 68Ga-pentixafor PET/CT in detecting CXCR4 expression levels in various LPDs, such as lymphoma and leukemia. However, the review could be enhanced by including more specific details on the sensitivity, specificity, and overall diagnostic accuracy of 68Ga-pentixafor PET/CT compared to other imaging modalities, providing a clearer assessment of its clinical utility. The review offers insights into CXCR4 expression patterns and their correlation with disease characteristics in LPDs.

It discusses how 68Ga-pentixafor PET/CT imaging can identify CXCR4-positive lesions and assess disease extent and aggressiveness. However, it would be beneficial to provide more in-depth analyses of the relationship between CXCR4 expression levels, disease subtypes, and clinical outcomes in different LPDs. Including this information would enhance the review's clinical relevance and applicability.

What about 18F- or 64Cu- labeled PET tracers?

Author Response

Response to Reviewer 1 Comments

1. Summary

  1. Point-by-point response to Comments and Suggestions for Authors

Comment 1: The literature review provides a comprehensive evaluation of [68Ga]Ga-Pentixafor PET/CT imaging in lymphoproliferative disorders (LPDs), demonstrating its potential as a valuable diagnostic tool. It effectively summarizes the utility of [68Ga]Ga-Pentixafor PET/CT in detecting CXCR4 expression levels in various LPDs, such as lymphoma and leukemia. However, the review could be enhanced by including more specific details on the sensitivity, specificity, and overall diagnostic accuracy of [68Ga]Ga-Pentixafor PET/CT compared to other imaging modalities, providing a clearer assessment of its clinical utility.

Response 1: Thank you for mentioning this point, we agree that comparison between different modalities used, such as PET/CT and (/MRI), can provide us with good diagnostic information, but this goal faces several limitations. First, in different lymphoproliferative studies, the number of studies in different modalities is either low or we only have one type of modality. In addition, few studies have reported the sensitivity and specificity of the modalities, and often the same evaluation index is not found in different groups, which is a requirement for scientific comparison.

This point was also added to the discussion.

Comment 2: The review offers insights into CXCR4 expression patterns and their correlation with disease characteristics in LPDs. It discusses how [68Ga]Ga-Pentixafor PET/CT imaging can identify CXCR4-positive lesions and assess disease extent and aggressiveness. However, it would be beneficial to provide more in-depth analyses of the relationship between CXCR4 expression levels, disease subtypes, and clinical outcomes in different LPDs. Including this information would enhance the review's clinical relevance and applicability.

What about 18F- or 64Cu- labeled PET tracers?

Response 2: Thank you. We also agree and the information on the expression level of different types of lymphoproliferative disorders can even help to distinguish them from each other, but only one study has been conducted in this regard, and even though we have added it in the results section, we cannot be sure decide about this.

About the comparison with other tracers such as FDG, considering that comparative studies with it were included in the study, based on your suggestion, we described the results of comparison with it in the results section in a separate microstructure.

Reviewer 2 Report

Comments and Suggestions for Authors

The introduction was very nicely done as was the review of each type of disease/publication of the evaluation of the radiotracer. 

As this is a review of multiple publications evaluating the performance of Ga-86 Pentixafor, some with straight comparisons to FDG, it might be nice to show the side by side comparisons. 

Author Response

Response to Reviewer 2 Comments

1. Summary

  1. Point-by-point response to Comments and Suggestions for Authors

Comment 1. The introduction was very nicely done as was the review of each type of disease/publication of the evaluation of the radiotracer. 

As this is a review of multiple publications evaluating the performance of [68Ga]Ga-Pentixafor, some with straight comparisons to FDG, it might be nice to show the side-by-side comparisons. 

Response 1: Thank you for mentioning this point, we agree and a separate microstructure was added in the results section for better clarity of the comparison results with FDG. The change of the results section was also applied according to this point.

Reviewer 3 Report

Comments and Suggestions for Authors

The authors performed a systematic review for a relevant clinical question. It is useful to have an overview of the literature for this topic.

The review was performed state of the art according to PRISM guidelines. The research was registered in PROSPERO. The research question in PROSPERO seems different from the submitted article and the registration was done late (search Dec 23, registration Feb 24).

Two authors performed the initial literature search, study selection, data extraction, and quality assessment. In addition, two authors revised and updated part of the literature search and study selection. More methodological details are needed. What did the authors exactly do. Were they independent? How were discrepancies handled? 

There are basically 3 questions: diagnostic accuracy, impact on staging, prediction of treatment response. It would help to add additional structure to the results section by adding the 3 components for each disease that is presented. 

Author Response

Response to Reviewer 3 Comments

1. Summary

  1. Point-by-point response to Comments and Suggestions for Authors

Comment 1: The authors performed a systematic review of a relevant clinical question. It is useful to have an overview of the literature on this topic.

The review was performed state of the art according to PRISM guidelines. The research was registered in PROSPERO. The research question in PROSPERO seems different from the submitted article and the registration was done late (search Dec 23, registration Feb 24).

Response 1: Thank you for pointing this out. Regarding the dates, we must state that the preliminary investigations to start the preliminary search were done in March 2023, but the registration on the PROSPRO website was done in January 2024, and while registering, we mentioned that the study is in the beginning stage, and ongoing. After registering and receiving the code, our search was updated again in February 2024.

Considering that the research question item is not one of the things that can be changed on the PROSPRO website, it was not possible to make it equal, however, the change request was registered on the site.

Comment 2: Two authors performed the initial literature search, study selection, data extraction, and quality assessment. In addition, two authors revised and updated part of the literature search and study selection. More methodological details are needed. What did the authors exactly do? Were they independent? How were discrepancies handled? 

Response 2: We agree, so the methodology of the study was explained in more detail and highlighted in the study.

“[Two people performed the initial search, which was done using the Endnote software (Version 21, USA) to accurately screen each title, abstract, and Full-text evaluation step. The steps of data extraction and study selection, data extraction, and quality assessment were carried out by the same two persons. The overlap between their results was 99%, and in a meeting regarding their difference, a conclusion was made based on the include and exclude criteria. Again, the search was revised by one person for the possibility of missing studies on the date (March 2024), and its results were added to the study]”

Comment 3: There are basically 3 questions: diagnostic accuracy, impact on staging, and prediction of treatment response. It would help to add additional structure to the results section by adding the 3 components for each disease that is presented. 

Response 3: Thank you for making this suggestion, we agree with it, so the microstructure of each of the evaluation results of lymphoproliferative types was divided into the mentioned sections including “diagnostic accuracy”, “impact on staging”, and “prediction of treatment response”.
